# Reinforcement Learning for Precision Oncology

**DOI:** 10.3390/cancers13184624

**Published:** 2021-09-15

**Authors:** Jan-Niklas Eckardt, Karsten Wendt, Martin Bornhäuser, Jan Moritz Middeke

**Affiliations:** 1Department of Internal Medicine I, University Hospital Carl Gustav Carus, 01307 Dresden, Germany; Martin.Bornhaeuser@uniklinikum-dresden.de (M.B.); janmoritz.middeke@uniklinikum-dresden.de (J.M.M.); 2Institute of Software and Multimedia Technology, Technical University Dresden, 01069 Dresden, Germany; karsten.wendt@tu-dresden.de; 3German Consortium for Translational Cancer Research, 69120 Heidelberg, Germany; 4National Center for Tumor Diseases, 01307 Dresden, Germany

**Keywords:** precision oncology, reinforcement learning, artificial intelligence, machine learning, dose adjustment, chemotherapy, radiotherapy

## Abstract

**Simple Summary:**

The accelerating merger of information technology and cancer research heralds the advent of novel methods and models for clinical decision making in oncology. Reinforcement learning—as one of the major subspecialties in machine learning—holds the potential for the development of high-performance decision support tools. However, many recent studies of reinforcement learning in oncology suffer from common shortcomings and pitfalls that need to be addressed for the development of safe, interpretable and reliable algorithms for future clinical practice.

**Abstract:**

Precision oncology is grounded in the increasing understanding of genetic and molecular mechanisms that underly malignant disease and offer different treatment pathways for the individual patient. The growing complexity of medical data has led to the implementation of machine learning techniques that are vastly applied for risk assessment and outcome prediction using either supervised or unsupervised learning. Still largely overlooked is reinforcement learning (RL) that addresses sequential tasks by exploring the underlying dynamics of an environment and shaping it by taking actions in order to maximize cumulative rewards over time, thereby achieving optimal long-term outcomes. Recent breakthroughs in RL demonstrated remarkable results in gameplay and autonomous driving, often achieving human-like or even superhuman performance. While this type of machine learning holds the potential to become a helpful decision support tool, it comes with a set of distinctive challenges that need to be addressed to ensure applicability, validity and safety. In this review, we highlight recent advances of RL focusing on studies in oncology and point out current challenges and pitfalls that need to be accounted for in future studies in order to successfully develop RL-based decision support systems for precision oncology.

## 1. Introduction

Recent advances both in terms of generating an ever-growing body of medical data and the increasing computational capacity to organize such data herald an accelerating merger of information technology and the medical domain. At the intersection of increasingly more complex medical data and computational analysis, machine learning (ML) gains a foothold driven by recent developments both in hardware components and accessible software technologies [1,2]. In general, machine learning encompasses three fundamental methodologies (Figure 1): supervised learning (SL), unsupervised learning (UL) and reinforcement learning (RL) [3].

In supervised learning, an algorithm is trained on a set of previously labeled data and learns features to map labels to unlabeled data of a test set. Ideally, it can then recognize and label real-world data, for example, for class prediction in histology, where it may distinguish between benign and malignant tissue. Another example is the detection of breast cancer in radiology, where pre-labeled images (benign/malignant based on histology) can be used to train an algorithm to spot malignancies and guide subsequent treatment planning [4,5,6,7,8,9]. In unsupervised learning, the data are unlabeled and clustered based on similarities and differences. This can, for example, be used to identify groups of patients at risk using genomics where different genetic clusters of a disease may have either favorable or unfavorable outcomes [10,11]. Both these methods are broadly applied to (often retrospectively) medical datasets and are utilized for diagnosis, risk stratification, genomic clustering, outcome prediction, relapse monitoring and treatment response prediction [12]. However, clinical practice is dynamic, and the question of how well algorithms that are exclusively trained on retrospective data perform in a prospective real-world setting remains unanswered in most cases. To address the challenge of a non-stationary clinical environment with changing conditions and stimuli, RL bears the potential to develop novel methods for data-driven computer-guided decision support systems. RL learns to select different actions according to different environmental states in order to maximize long-term rewards. This may be used for dynamic treatment regimens where doses are selected according to tumor and patient biology, treatment response and adverse events to tailor a treatment strategy that fits the individual patient.

In recent years, RL has rapidly evolved, demonstrating unprecedented success and often achieving human-level or superhuman-level performance in, for example, gameplay of complex board games such as chess, Shogi and Go [13,14,15], video games [16,17,18,19] and the field of autonomous driving [20].

Precision medicine aims at tailoring therapy and dosing to the individual patient based on individual intrinsic factors such as patient and disease biology that may affect the response to therapy, risk of treatment failure or relapse and prognosis [21]. Consequently, interventions can be adjusted to the individual patient or patient groups that may respond more favorably while, at the same time, reducing the risk of adverse events in patients who are unlikely to benefit. Both SL and UL currently receive the most attention as they offer insight into disease prognostication as well as treatment response using retrospective data. However, the dynamic situation both the individual patient and clinician find themselves in during oncologic treatment is not well captured by both SL and UL. The sequential foundation of RL provides a more suitable approach to capture the dynamics of oncologic therapy in a real-world (prospective) setting where both patient and environmental variables may change over the course of treatment.

In this review, we aim to provide a general understanding of the foundations of RL for the clinical oncologist, highlight previous studies of RL in oncology and outline potential pitfalls and considerations for future studies in this novel subfield at the intersection between healthcare and ML.

## 2. Overview of Reinforcement Learning

In this subsection, we provide a general overview of the concepts of reinforcement learning. We aim to inform the reader of the fundamental assumptions of RL, important terminology (Table 1) and different variations of RL methodologies (Table 2). For a more in-depth outline, we refer the interested reader to the detailed explanations provided by Sutton and Barto [22].

In RL, an agent interacts with its environment over time by selecting actions depending on the observed states of the environment while following a policy in order to maximize a cumulative reward (Figure 2) [22]. At each time point t, the agent observes a state st out of a pool of possible states S and selects an action from a pool of possible actions A following its policy π(at|st). For its choice of action according to the observed states of the environment, the agent receives a reward rt according to a reward function R and subsequently transitions to the next state st + 1 according to a transition function T. Finally, the return the agent receives is the accumulated reward discounted by the discount factor γ ∈ (0, 1] [23].

For example, an RL agent could be presented with multimodal patient data, e.g., demographics, laboratory values, tumor burden and therapy-associated toxicities, that represent the environment. For every iteration, the agent then selects an action, for example, a dose adjustment on a linear scale from 0 to 100%, given the state of the environment. This action will result in an alteration of the environment, i.e., of the patient’s condition and the data associated with it, resulting in a reward or a penalty for the agent based on whether or not the chosen action led to a favorable outcome for the patient. In that sense, the agent can abstract a policy either from rewards or state–action pairs that drives action selection, for example, the agent may learn that increasing doses of chemotherapy are associated with an increased anti-tumor effect, but also increased toxicity.

Hence, the agent ultimately aims at maximizing the long-term return from each state–action pair instead of short-term rewards by selecting the appropriate actions at each given state. If the problem setup enables an underlying model to be determined or learned from experience, for example, in a game setting with clearly defined rules and transitions, learning is referred to as model-based. If the model for state transition and reward is unknown, the agent learns directly from experience using a trial-and-error approach, and learning is referred to as model-free. Regarding nomenclature, if the underlying model is simulated while the data stem from a real-world cohort, the setup of the experiment is referred to as in virtuo [24]. However, if the data are simulated as well as the model, the experiment is referred to as in silico. In virtuo experiments are frequently performed when a retrospective patient cohort is available as a data source, while in silico experiments often require modeling of plausible real-world-like data, for example, the behavior of simulated cancer cells under the influence of chemotherapy. When the agent trains on data presented in a sequential manner, learning is referred to as online, while if all data are presented at the same time (i.e., in retrospective setups), learning is referred to as offline or batch mode [23]. If the entirety of the available action and observation space is known, a future state only depends on the current state and action (Markov criterion). This problem can be described as a Markov decision process (MDP) by a tuple of (S, A, R, T, γ) [22]. Receiving the maximum possible return in an MDP environment is achieved by optimizing the agent’s policy. For each policy π, a value function Vπ can be determined which predicts the expected reward the agent will accumulate when acting according to policy π in a state s [22]. As an alternative to the state value function, an action value function Qπ(s|a) can be determined that predicts the reward based on the agent taking a specific action a in a state s [22]. Both Vπ and Qπ can be expressed with the Bellman equation [25]. While both the value and policy iteration update all value states for each iteration, the temporal difference updates single state values for a given transition [22]. For example, in Q-learning, an estimate of the optimal action value function is updated at every state transition [26]. In most RL algorithms, approximations are usually presented in tabular form which may become problematic with high-dimensional data. The implementation of deep neural networks [27] to RL (deep reinforcement learning, DRL) does not require tabular representations for policies, value functions or Q. Recently, Mnih et al. [17] introduced deep Q-learning (DQL) that utilizes neural networks to directly learn policies from high-dimensional data and thereby overcomes the previous shortcomings of RL with neural nets by stabilizing the training of the action value function in an end-to-end RL approach while providing an algorithm that adapts to a variety of tasks (Atari games). More recently, Schrittwieser et al. [19] introduced MuZero that outperforms previous DRL algorithms in gameplay. In contrast to previous DRL algorithms, MuZero does not aim at modeling the entire environment but only models what is needed for the agent’s decision-making value, policy and reward—using a deep neural network and tree-based search.

These recent advances, especially in DRL, that are adaptive to an increasing range of settings without the need to fully disclose the underlying dynamics of an environment, i.e., the rules of the game, provide a vast potential for applications in oncology where an abundance of high-dimensional data and rapid environmental changes limited previous efforts.

## 3. Recent Studies of Reinforcement Learning in Malignant Disease

Treatment regimens in oncology are usually longitudinal decision-making processes where patient variables as well as response to treatment and toxicities influence the oncologist’s choice in order to optimize patient safety and outcome in the long run. This clinical framework can be translated into a set of sequential actions in an environment that result in iterative state alterations. In that sense, dynamic treatment regimens (DTRs) [28] can be set up as an RL problem due to its sequential nature, and dose adjustments can be performed by a digital agent that receives rewards for favorable events such as tumor response or curation and penalties for unfavorable events such as toxicities (Figure 3) [29]. Due to obvious ethical concerns in a trial-and-error learning method, RL in DTRs is usually applied, at present, in a retrospective setting or with simulated data based on historical cohorts.

Several recent studies have applied this approach for optimizing chemotherapy dosages, most commonly using Q-learning in simulated environments (Table 3). Padmanabhan et al. [30] employed Q-learning for chemotherapy dosing in an in silico approach in simulated patients in a closed loop to maximize on-target drug effects and minimize off-target toxicities. Additionally, utilizing Q-learning, Zade et al. [31] proposed a simulation framework where an RL agent optimizes the dosage of temozolomide in order to minimize glioblastoma tumor size. Yazdjerdi et al. [32] applied Q-learning to optimize anti-angiogenic therapy in a simulated tumor environment. RL-based drug sensitivity screening regarding different tumor cell lines with Q-rank was proposed by Daoud et al. [33]. Their method ranks drug sensitivity prediction algorithms and recommends the optimal algorithms for a given drug–cell line pair in order to achieve optimal responses. To account for chemotherapy-associated toxicity, Maier et al. [34] proposed an RL-based framework that is guided by absolute neutrophil counts for adjusting subsequent drug doses. Using simulated reinforcement trials [35], Zhao et al. [36] applied Q-learning to stage IIIB/IV non-small cell lung cancer and reported optimized first and second treatment lines as well as optimal selection for initiating second-line therapy. Similarly, Yauney et al. [37] aimed to minimize mean tumor diameters in a simulated trial of patients receiving chemo- and/or radiotherapy using action-derived rewards as approximations of patient outcome. Both Liu et al. [38] and Krakow et al. [39] used registry data from patients with hematologic malignancies who underwent allogeneic stem cell transplantation that were listed in the Center for International Blood and Marrow Transplant Research registry and applied DRL and Q-learning, respectively, in order to prevent and treat graft-versus-host disease.

In parallel to chemotherapy regimens, RL can be applied to optimize radiotherapy to maximize on-target effects and minimize off-target toxicities (Table 4). Treatment planning and manual target segmentation still require an excessive amount of manual labor and time [40]. Deep learning has been investigated in order to aid the radiotherapist in treatment planning and reduce inter-observer variability. For example, different variations of convolutional neural nets have been developed for fast and accurate segmentation of brain metastases [41,42], thoracic cancer manifestations [43] or rectal cancer [44]. Accordingly, RL can be used for radiation dose adjustments for the individual patient. Kim et al. [45] defined the radiotherapeutic fractionation schedule as an MDP and proposed adaptive fractions according to individual patient response. Tseng et al. [46] used deep Q-learning to develop adaptive radiation protocols for patients with non-small cell lung cancer, balancing rewards for the agent between on-target efficiency and off-target toxicity. They accommodated for the initially small sample size with simulated patient data generated by a generative adversarial net (GAN). Jalalimanesh et al. [47] used an agent-based model and Q-learning to adapt fraction sizes to tumor response in a simulated environment. Similarly, adjustment of dose fractionation performed by a DRL agent in a simulated model of tumor growth was also demonstrated by Moreau et al. [48], who reported an improved performance compared to the baseline treatment plans. Using historic data from prostate cancer patients, Hrinivich et al. [49] applied deep Q-learning for volumetric modulated arc therapy and reported on-target and off-target doses comparable to clinical plans. Correspondingly, Shen et al. [50] used DRL in a virtual environment to generate treatment plans by training on 10 and validating on 64 cases of patients with prostate cancer. In a similar approach, Zhang et al. [51] trained an RL agent on augmented treatment plans of 16 previously treated patients that received pancreas stereotactic body radiation therapy which was validated on 24 treatment plans, achieving a treatment quality comparable to clinical plans. It is to be noted that while the majority of the presented studies describe their algorithms in great mathematical detail, the information about the general problem setup and algorithm architecture has to be easily accessible to both software engineers and clinicians. In order to transparently report the used methodologies, authors of future studies of RL in medicine can refer to the proposed items in Table 2 for preparation of their paper’s method section. This framework can help both the authors in clearly structuring their reports and the readers in effortlessly picking out the main components of a novel algorithm architecture for a given use case. Using such a standardized approach can help facilitate the reproducibility of RL research in medicine and may aid in transferring RL algorithms from one application in oncology to another.

## 4. Discussion

The presented studies underline the feasibility of RL-guided precision oncology both regarding irradiation and drug therapy. However, the majority of previous studies suffer from common obstacles. In this section, we aim to highlight frequent challenges in RL design for clinical use cases and discuss possible strategies to overcome these hurdles. Accurately mapping the environment and assessing the complexity of available data as well as the sequential nature of a clinical problem are the key first steps in setting up an RL support system. Biological systems and their behavior under environmental influences represent a highly complex system with a myriad of unknown variables in the context of disease and treatment. Hence, a detailed model of a clinical problem can often not be obtained. Several studies address this issue by using simplified simulations of tumor behavior [30,31,32,36,37,46,47]; however, the majority of these studies worked with relatively small samples which can limit the agent’s capability of abstracting an efficient policy given few examples to train on [52]. This leads to the question of to what extent such algorithms are generalizable to more complex environments or real-life applications. Sparse and missing data are all too common in medical datasets. If the agent cannot access all information that is critical for decision making, a concluding model may misrepresent the actual environment. In that sense, most scenarios in clinical oncology behave in a non-Markovian way as not all relevant information is disclosed to the agent (or the clinician) [53]. Adding to the complexity, medical data may be biased or noisy due to inter-rater variability depending on the data source which may add to the variance of estimates of the value function and therefore affect policy determination [29]. While many cases of missing data in RL in general may be tackled with a partially observable Markov decision process design [54], the high dimensionality and complexity of medical data demand more sophisticated methods, such as multiple imputation models [55] or advanced Q-learning techniques for patients lost to follow-up [56]. Small sample sizes can be accounted for by pooling multicenter datasets which may, in turn, add variability to the dataset. Hence, standardization of data collection across institutions and even countries seems warranted to generate large high-quality datasets for future ML applications. To maintain high quality in such multicenter and multinational datasets, standardization of reporting as well as public access is essential. Internationally acclaimed frameworks for tumor response such as RECIST [57] or the reporting of adverse events such as CTCAE [58] as well as data from electronic health records [59] can be utilized to store clinical information in such datasets in a universally accessible way without the need for excessive pre-processing before pooling data from different sources. Data sharing between institutions and countries is crucial to create larger datasets, even for rare entities, and provide RL agents (and ML in general) with bigger sample sizes to train on. A frequent shortcoming of the studies cited above (with a few exceptions) is the lack of publicly available datasets and code. Often, only mathematical modeling or pseudocode is reported. However, to ensure reproducibility, public availability of both data and code is key. This will allow for independent model improvement, pooling of similar datasets and, overall, a faster pace and higher generalizability of RL models in oncology. Publishers should acknowledge this shortcoming and incentivize authors to share their code upon publication. However, informed patient consent about the processing of data and safety measures to protect patient identity need to be implemented. As this process naturally requires collaborative efforts and time, alternative approaches are needed for small data. GANs [60] can be implemented to augment small datasets. Their feasibility to add data to RL has recently been demonstrated in a dataset of patients with non-small cell lung cancer [46]. However, a study evaluating RL performance in a comparison between real-world and simulated data is lacking. Such a comparison could be made between a dataset generated by GANs and retrospective patient data in order to show discrepancies based on the data structure. This would allow for an in-depth look at the quality of simulated data which, in turn, could be improved to allow for a more robust simulation in future models. Another possibility is to first train the RL agent by expert demonstration, inverse learning, transfer learning or a combination thereof. Formulating a reward function a priori and then letting the agent derive an optimal policy may not be ubiquitously possible in a clinical setting with many unknown variables, and hence retrospective data of (near-)optimal treatment histories can be utilized to estimate a reward function based on previous expert decisions [29]. This can be achieved by behavioral cloning, where pairs of environmental states and expert actions are mapped directly by the agent [61,62] (in a way similar to supervised learning), or by inverse RL, where a reward function is determined based on observing ideal decisions [63,64]. Still, it needs to be considered that in this scenario, the reward is based on a match between the agent’s and the expert’s decision, which may, again, result in bias as different experts may disagree over different decisions, and therefore misrepresentative rewards can result in poor performance and safety issues [65]. This leads to a fundamental issue at the heart of RL: credit assignment. The main incentive to reinforce an agent’s behavior is encoded in the reward function, and henceforth, the reward signal determines whether or not a certain behavior is reinforced given a certain state of the environment. While this may be straightforward in gameplay where all underlying dynamics are known and the reward is often a direct consequence of an agent’s action, rewards in a healthcare domain may be sparse, and the time between an action and its result may be considerably longer. In oncology, treatment effects evidently do not manifest themselves immediately, and linking an agent’s action, e.g., a dose modification, to a certain outcome, e.g., prolonged relapse-free survival, remains challenging. Consequently, long-term rewards should be favored over short-term rewards, and oversimplifying reward functions can lead to unwanted behaviors, resulting in an agent doing more harm than good [52]. Furthermore, in comparison to gameplay where there usually is one single goal (win the game), oncologic practice demands a variety of treatment goals to be met such as improving survival, reducing morbidity, reducing toxicity and improving quality of life, among others. A possible way to deal with sparse rewards in the light of multiple goals is hindsight experience replay, where different learning episodes are replayed with different goals and the agent can derive reward signals regarding different outcomes [66]. In most applications of RL in healthcare, rewards are coded quantitatively rather than qualitatively, which can be useful for certain use cases where the outcome, in fact, is a metric variable (such as absolute neutrophile count [34]); however, it remains challenging when the outcome first has to be transformed or a priori model building has to be performed manually [29]. Alternatively, preference models can be used as a representation of qualitative feedback to rank the agent’s behavioral trajectories [67,68]. However, a critical question is whether the reward an agent receives for an action is actually the optimal possible reward. This leads to another fundamental issue in RL, the trade-off between exploitation and exploration. Essentially, an agent has two options: either exploit current knowledge in order to achieve rewards or explore for previously unknown information which potentially leads to improved policies to gain higher rewards [22]. In the healthcare domain, especially in oncology, this dualism is crucial since exploration methods with insufficient safety measures can lead to potentially devastating outcomes, while insufficient exploration may lead to suboptimal policies and thus to unsatisfactory treatment decisions. Penalizing an agent for an unfavorable action may be insufficient when it comes to safety concerns in a healthcare setting, especially when the action leads to unrecoverable damage. This becomes especially relevant when dealing with drugs that have narrow therapeutic ranges and information on dose adaptation is limited [69]. Adding to the aforementioned challenge of multiple objectives in a healthcare setting is the fact that some objectives may be contradictory. For example, a full dose of chemotherapy may result in improved tumor response but, at the same time, will inevitably increase toxicity. A method to account for such contradictions is multi-objective RL that aims to evaluate polar objectives by obtaining a policy that represents Pareto optimal solutions [70]. While a lost game can simply be reset and started anew, an overdose in a clinical setting can potentially cost a patient’s life. Hence, safe exploration strategies [71], especially in online learning, are crucial for RL in oncology. This raises the question of what the optimal benchmark should be when it comes to evaluating an RL agent’s performance. Frequently, RL decisions are compared to clinical treatment plans; however, it remains questionable whether this is the optimal strategy since, conceivably, RL performance in a narrow domain could, at some point, exceed human performance in terms of decision making as it already has done, for example, in chess. When it comes to decision support systems, safety goes hand in hand with trust. Let us assume that your RL algorithm suggests a dose alteration for a given patient. Do you trust that decision? If so, why? If not, why not? A major drawback of many current ML applications in such delicate environments as healthcare is interpretability, and DL in particular is often referred to as a ‘black box’ when it comes to exactly how an algorithm arrives at a conclusion [72,73]. This becomes especially challenging when the oncologic expert and the RL algorithm arrive at different solutions for the same problem [74,75]. Often, the signals an algorithm uses for decision making and the policies that are learnt can neither be accessed easily nor interpreted comprehensively by the human investigator [76]. Yet, the path to the conclusion is equally as important as the conclusion itself, especially in healthcare, where not only scientific knowledge gains are expected but patients also have an inherent right to be well-informed with respect to the background of a treatment decision. The interpretability of such RL algorithms should refrain from unnecessary abstraction and highlight causal pathways [77] that are meaningful to both the clinician and the patient. In that sense, understanding the exact model may be unnecessary in practice (to the clinician and patient) when causal pathways can be well interpreted. However, this is still an ongoing endeavor in ML in general [78,79], and satisfactory solutions tailored for healthcare applications are lacking [80], which bears the risk of reintroducing a paternalistic system in patient care [81]. Still, this remains controversial as it can be argued that the input from ‘black box’ systems is already happening to some extent in clinical oncology and is widely accepted in daily practice: hardly anyone seriously questions the results of molecular analysis or the assessment of biomarkers when it comes to clinical decision support, and confidence in these techniques has been built over recent years by reliable performance [82]. It is therefore conceivable that RL-based decision support systems, once they have been broadly tested and validated, may gain a similar level of trust as advanced biomedical techniques. In that regard, the frequent notion that ML systems could threaten the clinician’s autonomy can be set aside as it is far more likely that these systems will be integrated as decision support in the same way that molecular and genetic data are implemented now, guiding precision oncology and further individualizing patient care, while the final responsibility for any taken decision undoubtedly lies with the oncologist. Previous studies focused on either conventional chemotherapy regimens or radiotherapy. However, the implementation of targeted therapy or immunotherapy in the treatment guidelines of many tumor entities calls for studies that account for these therapeutics as well and evaluate combinations of chemo-, radio- and targeted therapy in the respective tumor entities. These studies and algorithms have to be designed with diligence to both accurately map a clinically relevant problem setup in oncology and, at the same time, account for multiple different objectives and potential adverse effects in the context of multimodal contemporary therapy regimens.

## 5. Conclusions

RL in oncology is still in its infancy, and as we pointed out, a multitude of issues have to be properly addressed in future studies for these techniques to mature and find acceptance in clinical oncology. The sequential nature of RL and its capability for long-term outcome optimization make it a suitable candidate to be implemented in precision oncology, harnessing the growing body of available biomedical data for the individual patient. To progress in this potentially practice-changing field, an interdisciplinary effort to iteratively refine these systems for specific use cases as well as institutional guidelines is needed in order to achieve meaningful representations of clinically relevant tasks for optimal patient care.

## Figures and Tables

**Figure 1 cancers-13-04624-f001:**
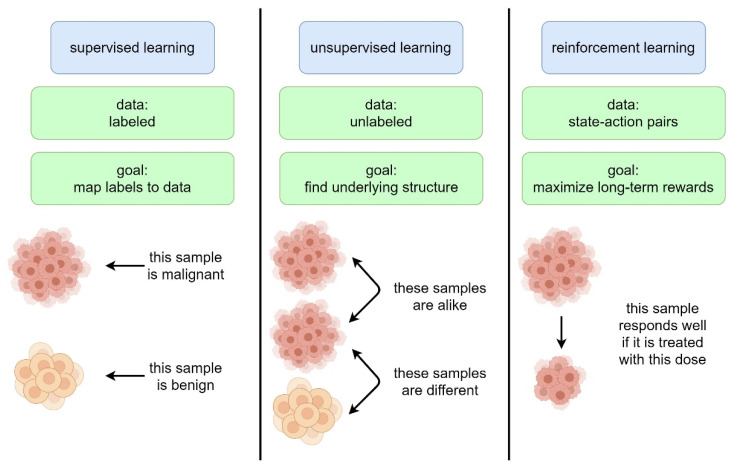
Main differences between machine learning techniques.

**Figure 2 cancers-13-04624-f002:**
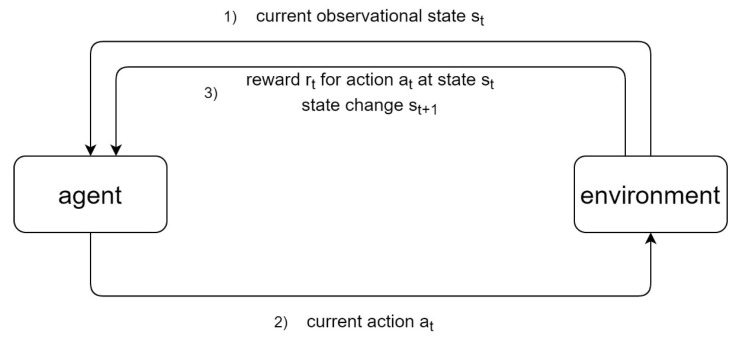
Concept of reinforcement learning. An agent receives current state observations from an environment (1) and, in response, selects an action according to its policy (2). For this action, the agent receives a reward based on a reward function and the state of the environment changes (3). The agent’s goal is to maximize long-term rewards and achieve the optimum possible return.

**Figure 3 cancers-13-04624-f003:**
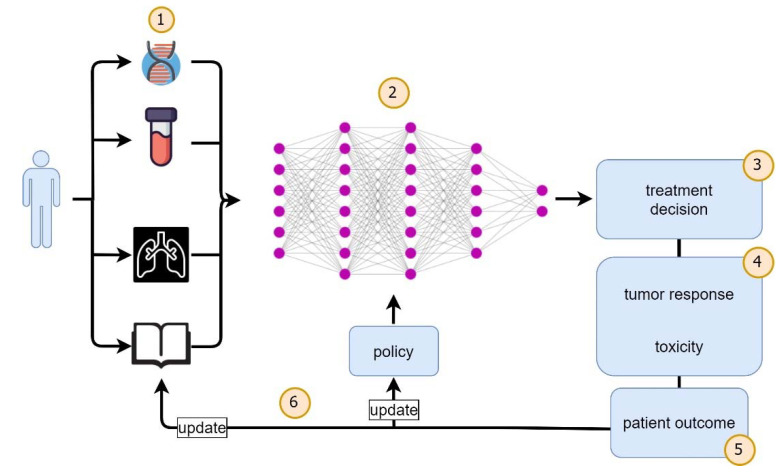
Iterative workflow of a reinforcement learning approach to precision oncology. For the individual patient, multimodal data (1), e.g., from genetic assays, laboratory tests, radiographic images and electronic health records, serve as an input for a reinforcement learning (RL) framework (2)—here, depicted as a deep neural net—which selects an action such as a treatment decision (3) according to its policy. This treatment decision will affect tumor response and toxicity (4) simultaneously and thus, ultimately, affect long-term patient outcome (5). This is translated into a reward signal for the RL agent which results in a policy update (6). At the same time, the state of the patient changes. For example, tumor response could be measured with radiographic imaging, and/or toxicity could be monitored by laboratory values and documentation in the electronic health record. This update of the state initiates a new cycle where updated inputs to the RL framework lead to a new treatment decision according to the updated policy, and the loop is closed.

**Table 1 cancers-13-04624-t001:** Terminology of reinforcement learning.

Term	Symbol	Description
Reinforcement Learning	RL	operates in a simulated environment with distinct behavior to receive rewards
Environment	E	consumes actions to produce rewards for an agent; based on a model/simulation/observations/data
Agent		RL decision instance, performing actions to change states
Action	a	performed by an agent to change to another state, i.e., interact with the environment
State	s	abstract relation of the agent to the environment, starting and end point of an action
Reward	r	gain for an action of the access of a state
Reward Function	R	entirety of all rewards for actions/states
Cumulative Reward	CR	aggregated rewards of subsequent actions/states; should be maximized as the learning/optimization objective
Policy	π	defines an action for each state; result of learning/optimization process

**Table 2 cancers-13-04624-t002:** Variants and methodologies of reinforcement learning.

Aspect	Variant	Description	Pro	Contra
Environment	Model-Based	distinct rule-based/simulation-based feedback for the agent	covers corner cases, potentially high feedback quality	complex to set up
	Model-Free	data-based (observation/retro-perspective) feedback	easy to set up, no abstraction	no corner cases, potentially low feedback quality
Reward	V (State-Based)	rewards when accessing a state (relation to E)	fewer states, easy to model	more abstraction, static (less intuitive) view
	Q (Action-Based)	rewards when executing an action (changing E)	more actions, fewer abstraction, extensive to model	more actions, dynamic (intuitive) view
	Concluding Learning	rewards when finalizing a sequence of decisions	long term-oriented, aims for global objectives	provides no local guidance, complex evaluation
	Temporal Difference Learning	rewards after each decision	provides no local guidance, easy evaluation	short term-oriented, aims for local objectives
Access	Online	access of the agent to the E in a (restricted) stream-based way	less information to process for the agent, smaller solution space	potentially non-optimal solutions (policy)
	Batch-Based	access of the agent to the entire environment E	globally optimized solutions (policy)	more information to process, large solution space
Dynamics	Static Reward Function	each piece of feedback from the E is encoded in states, resulting in constant rewards	easier E, smaller solution space	potentially coarse-grained decisions/optimization
	Dynamic Reward Function	feedback from E is encoded in attributes, resulting in variable rewards	potentially fine-grained decisions/optimization	complex E, large solution space
	Markov Assumption	no influence from previous decisions	smaller solution space	potentially insufficient decision impact modeling
	No Markov Assumption	decision history has influence on rewards	complex decision modeling	large solution space
Representation	Table-/Map-Based	simple state transition/action modeling	easy to create, transparent	complex to maintain and show, grows exponentially with number of states
	Graph-Based	intuitive state machine modeling	easy to maintain, transparent, scales with number of states	complex to create and show
	Deep Neural Net	DL-based modeling	easy to create, scales with number of states	low transparency, complex to show

**Table 3 cancers-13-04624-t003:** Recent studies of reinforcement learning (RL) for adaptive dosing of antineoplastic drugs in cancer.

Reference	Main Goal	Environment/Cohort	Model-Based	Model-Free	V (State-Based)	Q (Action-Based)	Markov Assumption	No Markov Assumption	Table-/Map-Based	Deep Learning	Code Availability
[30]	Evaluation of an RL-based drug controller to enhance therapeutic effect on simulated tumors while sparing normal tissue without the necessity to disclose underlying system dynamics to the RL agent	15 simulated cancer patients		X		X	X		X		
[31]	Comparison of an RL-guided temozolomide treatment schedule to conventional clinical regimen	simulated glioblastoma tumor growth model	X			X	X		X		
[32]	RL-based optimization of anti-angiogenic therapy with endostatin in a simulated tumor growth model with dynamic patient parameters	simulated tumor growth model, simulated patient		X		X	X		X		X
[33]	Prediction of chemotherapy sensitivity in breast cancer cell lines with available multi-omics data by ranking suitable prediction algorithms using Q-rank	drug sensitivity data of 53 breast cancer cell lines	X			X	X		X		X
[34]	Evaluation of data assimilation techniques in combination with RL for dose adjustments of chemotherapy in simulated patients using absolute neutrophile count as a surrogate endpoint	simulated patients	X		X		X		X		X
[36]	RL-based dose adjustments for chemotherapy and initiation of second-line therapy while accounting for patient censoring	simulated clinical trial of stage IIIB/IV non-small cell lung cancer patients		X		X		X	X		
[37]	Deep RL-guided dosing regimens with temozolomide or procarbazine, CCNU and vincristine using action-derived rewards	simulated clinical trial using a glioblastoma tumor growth model	X			X	X		X		
[38]	Evaluation of RL-guided prevention and treatment of acute and chronic graft-versus-host disease	registry data from 6021 AML patients who underwent allogeneic stem cell transplantation		X		X		X		X	
[39]	Evaluation of RL-guided prevention and treatment of acute and chronic graft-versus-host disease	registry data from 11,141 patients who underwent allogeneic stem cell transplantation	X			X		X	X		

**Table 4 cancers-13-04624-t004:** Recent studies of reinforcement learning (RL) for adaptive dosing and fractionation of radiotherapy in cancer.

Reference	Main Goal	Environment/Cohort	Model-Based	Model-Free	V (State-Based)	Q (Action-Based)	Markov Assumption	No Markov Assumption	Table-/Map-Based	Deep Learning	Code Availability
[45]	Development of adaptive fractionation schemes based on mathematical modeling with a Markov decision process	Simulated environment of target volumes and organs at risk	X		X		X		X		
[46]	Evaluation of a multi-step deep learning model for radiation dose adjustments in a retrospective and augmented patient cohort compared to clinical treatment plans	Retrospective data of 114 non-small cell lung cancer patients and augmented data from a generative adversarial net		X		X		X		X	
[47]	Proof of concept of an RL agent for adaptive irradiation dosing and fractionation schemes	Simulated tumor growth model		X		X		X	X		
[48]	Comparison of adaptive dose fractionation schemes to clinical treatment regimens	Simulated tumor growth model	X			X	X			X	X
[49]	RL to guide volumetric modulated arc therapy with machine parameter optimization and comparison between on-target and off-target doses	Retrospective data of 40 patients with prostate cancer	X			X		X		X	
[50]	Training and evaluation of a RL-based deep virtual treatment planner	Retrospective data of 74 patients with prostate cancer		X		X		X		X	
[51]	Optimization of on-target and off-target dosing for stereotactic body irradiation in pancreatic cancer	Retrospective data of 16 patients with pancreatic cancer		X		X		X	X

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
