# Peer review of "Reinforcement Learning for Precision Oncology"

_cancers, 2021, doi:10.3390/cancers13184624_

Round 1
Reviewer 1 Report
The authors highlighted recent advances of Reinforcement Learning (RL) focusing on studies in oncology and pointed out current challenges and pitfalls that need to be accounted for in future studies in order to successfully develop RL-based decision support systems for precision oncology.
Overall, it is a very detailed and good-written review and then appropriate for being published on this special issue.
Some minor remarks are reported below:
- The authors stated ‘This clinical framework can be translated into a set of sequential actions in an in silico environment that result in iterative state alterations’(line 148). According to the work of Travassos, Guilherme Horta, and Márcio O. Barros, entitled "Contributions of in virtuo and in silico experiments for the future of empirical studies in software engineering.", simulated experiments can be classified as in virtuo and in silico While the in silicoexperiments simulates both the data and the model, the in virtuo ones simulates only the model. Please specify when it is suitable to apply RL for in virtuo or in silico experiments.
- According to the authors statement: “ It is to be noted that while the majority of the presented studies describes their algorithms in great mathematical detail, the information about general problem set-up and algorithm architecture has to be easily accessible to both software engineers as well as clinicians.”(line 216), I suggest to insert in tables III and IV one column referred to software availability since nowadays open source software-availability could improve the repeatability of the proposed setting.
- Are there any studies in literature that make a comparison among RL models applied on simulated data and on real clinical data? How could we measure the reliability of these models on clinical practice?
- Please include some references related to (i) SL and (ii) UL algorithms for precision oncology in the Introduction such as:
(i) RodrÃguez-Ruiz, A. et al. Detection of breast cancer with mammography: Effect of an artificial intelligence support system. Radiology 2018, 290, 305–314.
Perek, S. et al. Classification of contrast-enhanced spectral mammography (CESM) images. Int. J. Comput. Assist. Radiol. Surg. 2018, 14, 1–9.
Massafra,R. et al. Radiomic Feature Reduction Approach to Predict Breast Cancer by Contrast-Enhanced Spectral Mammography Images. Diagnostics 2021,11,684.
(ii) Amoroso, Nicola, et al. "A Roadmap towards Breast Cancer Therapies Supported by Explainable Artificial Intelligence." Applied Sciences11 (2021): 4881.
Jafari, Mohieddin, et al. "Unsupervised learning and multipartite network models: a promising approach for understanding traditional medicine." Frontiers in pharmacology 11 (2020): 1319.
Author Response
We want to thank the reviewer for the insightful remarks and detailed commentary. The distinction between in virtuo and in silico experiments is indeed a very important one, especially in the context of model-based RL. While technically many studies so far use in virtuo approaches as the data is based on retrospective cohorts and only the model is simulated or approximated as long as model-based RL is used, there are some studies such as Tseng et al. that also simulate the data (at least partially) using generative adversarial nets. This qualifies as a classical in silico approach as defined by Travassos et al. We agree that this distinction is important for the reader to understand different architectures of model-based approaches and use of data in such models. We added:
“Regarding nomenclature, if the underlying model is simulated while the data stems from a real-world cohort, the set-up of the experiment is referred to as in virtuo [Travassos and Barros, 2004]. However, if the data is simulated as well as the model, the experiment is referred to as in silico. In virtuo experiments are frequently performed when a retrospective patient cohort is available as a data source, while in silico experiments often require modeling of plausible real-world-like data, for example the behavior of simulated cancer cells under the influence of chemotherapy.” Lines 129-135
We also want to thank the reviewer for the excellent suggestion regarding code availability. We agree that sharing data and code is essential to transparently communicate scientific results in information technology and to ensure reproducibility. We added this information to tables 3 and 4. Unfortunately, the majority of cited articles do not publicly share their code. We believe this is an urgent issue worthy of discussion and to raise awareness, we added the following statement to the discussion section:
“A frequent shortcoming of the studies cited above (with a few exceptions) is the lack of publicly available data sets and code. Often, only mathematical modeling or pseudocode is reported. However, to ensure reproducibility, public availability of both data and code is key. This will allow for independent model improvement, pooling of similar data sets and overall a faster pace and higher generalizability of RL models in oncology. Publisher should acknowledge this shortcoming and incentivize authors to share their code upon publication.” Lines 286-292
Regarding a comparison between RL on simulated and real data, we have to acknowledge that the overall number of studies using RL in the context of oncology is small so far. The only study we found that explicitly uses both real-world data from a retrospective cohort and at the same time simulated data from generative adversarial nets (GAN) is Tseng et al., however, they do not explicitly compare RL performance in terms of data source. GANs were used simply to increase sample size for training. We absolutely agree with you that such a comparison is very interesting and, in fact, we are currently working on such a GAN-based comparison of data sources between real and simulated data, however, with a supervised learning framework rather than an RL framework. We added:
“However, a study evaluating RL performance in comparison between real-world and simulated data is lacking. Such a comparison could be made between a data set generated by GANs and retrospective patient data in order to show discrepancies based on data structure. This would allow for an in-depth look at the quality of simulated data which in turn could be improved to allow for more robust simulation in future models.” Lines 297-302
With respect to reliability of these models, we already discuss hurdles such as credit assignment and acceptance of RL in clinical practice. However, a universal measurement of reliability is hard to conceive. Several mentioned studies (Hrinivich et al., Shen et al., Zhang et al.) use comparisons to clinical treatment plans as a measurement, however, we believe this is controversial. Potentially, RL performance could exceed clinicians’ performance in radiotherapeutic treatment planning as it did in chess, for example. Using human-generated benchmarks therefore may not be optimal as a measurement of success. Ultimately, prospective studies are needed to demonstrate superiority (or at least equivalence) of RL-based treatment plans and adjustments in comparison to human-generated regimens, however, as we already discuss in-depth, this is ethically questionable. We added:
“This raises the question of what the optimal benchmark should be when it comes to evaluating an RL agent’s performance. Frequently, RL decisions are compared to clinical treatment plans, however it remains questionable whether this is the optimal strategy since conceivably RL performance in a narrow domain could at some point exceed human performance in terms of decision making as it already did for example in chess.” Lines 356-361
We agree with the reviewer that supervised and unsupervised learning have already demonstrated tremendous progress in precision medicine, especially in, for example, breast cancer detection and we thank the reviewer for the excellent suggestions regarding literature that underlines this progress and included all these studies as examples for SL and UL.
Reviewer 2 Report
The authors discuss a topic of particular interest for the scientific community, we highlight the recent advances in Reinforcement learning in oncology and the current issues that need particular attention in the development of decision support systems based on RL. The manuscript is generally well written and structured. However, here are some suggestions.
1) I would suggest restructuring the introduction: the description of supervised and unsupervised techniques are introduced from line 43 and then resumed again at line 54 for each of them. Perhaps it would be more linear to describe each of them in a single solution. Then describe the Reinforcement learning with respect to these in general and its use in clinical practice.
2) some recent works of clinical interest should be introduced:
Comes, Maria Colomba, et al. "Early Prediction of Breast Cancer Recurrence for Patients Treated with Neoadjuvant Chemotherapy: A Transfer Learning Approach on DCE-MRIs." Cancers 13.10 (2021): 2298.
Amoroso, Nicola, et al. "A Roadmap towards Breast Cancer Therapies Supported by Explainable Artificial Intelligence." Applied Sciences 11.11 (2021): 4881.
Dembrower, Karin, et al. "Effect of artificial intelligence-based triaging of breast cancer screening mammograms on cancer detection and radiologist workload: a retrospective simulation study." The Lancet Digital Health 2.9 (2020): e468-e474.
3) Paragraph 2 "Overview of Reinforcement Learning" is at times too technical for cutting the article whose purpose seems to be of interest in the usefulness in clinical practice of approaches based on such artificial intelligence techniques.
Author Response
We want to thank the reviewer for the insightful suggestions. We agree that the previous structure of the introduction could be improved. We re-structured the introduction according to the reviewer’s suggestion and explain supervised learning, unsupervised learning and reinforcement learning in a more linear manner and only later on highlight the advantages of RL in precision medicine.
We also agree that recent advances in the detection of breast cancer are a beacon for artificial intelligence in medicine. This is highlighted by the studies suggested by the reviewer which are a prime example how AI can support cancer diagnosis and subsequent treatment. We added breast cancer as an example of this progress of AI in medicine in the introduction and included all studies mentioned by the reviewer as references for the readership.
Regarding paragraph 2, we agree that it is technical, however, we strongly believe that it is necessary for the reader to be provided with the essentials of RL in order to understand the at times much more complex models provided by the studies referenced in the following paragraph. Since the technical terms can be too abstract at times, we provided a clinical example in paragraph 2 that can be applied to the basic principles of many of the referenced studies. We added: “For example, an RL agent could be presented with multimodal patient data, e. g. demographics, laboratory values, tumor burden and therapy-associated toxicities, that represent the environment. For every iteration, the agent then selects an action, for example a dose adjustment on a linear scale from 0-100% given the state of the environment. This action will result in an alteration of the environment, i. e. of the patient’s condition and the data associated with it, resulting in a reward or a penalty for the agent based on whether or not the chosen action led to a favorable outcome for the patient. In that sense, the agent can abstract a policy either from rewards or state-action pairs that drives action selection, for example the agent may learn that increasing doses of chemotherapy are associated with increased anti-tumor affect, but also increased toxicity.” (Lines 113-122). This general framework is the underlying principle of most of the referenced literature. We believe this may aid the reader in better understanding the basic vocabulary of RL, however, as most of the literature regarding RL uses the described vocabulary, we do not see a benefit in further abstraction since general concepts should be transparent to the reader regardless of which of the references he/she refers to for further inspection.
Reviewer 3 Report
The manuscript is well written and have added a lot of knowledge to the field. However, I am unsure of what audience is targeted. I suppose the audience should be scientists or clinicians working in oncology and wanting to use A.I to aid in patient care. However, the whole paper would be very difficult to understand for scientists and clinicians and not enough examples are given which will give the content some context in the area of patient care.
For example, all the terminology has to be explained to a clinician audience, words like 'environment' 'agent' 'reward' 'state' etc - they have to be given a medical/clinical context and examples must be given.
The passages from line 107 to 139 are too confusing and I found it difficult to follow and not able to apply the knowledge to clinical work. This also applies to the passages from line 250 to 280.
The passages from line 293 is well explained and interesting to the clinician.
Overall, the paper is very interesting and attempts to explain the use or mis-use of A.I in the clinical arena, however, some editing is needed before publication.
Author Response
We want to thank the reviewer for the insightful comments. We agree that the topic is complex for a clinical readership. However, we strongly believe that the appropriate nomenclature has to be used since these vocabularies are universal in reinforcement learning and any further abstraction may not be beneficial to the readership. Seemingly, the reviewer is concerned with the readers’ understanding of paragraph 2 “Overview of Reinforcement Learning”. First, we have to acknowledge that this part of the manuscript already encompasses information from the cited references that is limited to the bare essentials of RL. We often left out extensive explanations of, for example, how exactly a value function is determined or how credit assignment is implemented as we deemed it unnecessary to extensively explain this to a primarily clinical and biomedical readership. Nevertheless, the presented information is the very foundation that studies of RL are based on and dropping any of the mentioned concepts would in our view hinder the reader from understanding the frameworks the cited studies of RL in oncology used. Our goal with this review is to provide a framework for future studies of RL in oncology aiming at clinicians, biomedical researchers and information technologists. We aim to incentivize collaboration of these disciplines. To do so, we deem it necessary to explain the fundamental assumptions of RL, so that each discipline is on the same page. The discussion therefore, as the reviewer has remarked, primarily aims at clinicians who may ultimately implement such algorithms in the future.
However, we agree with the reviewer that these concepts need clinical examples. We therefore included: “For example, an RL agent could be presented with multimodal patient data, e. g. demographics, laboratory values, tumor burden and therapy-associated toxicities, that represent the environment. For every iteration, the agent then selects an action, for example a dose adjustment on a linear scale from 0-100% given the state of the environment. This action will result in an alteration of the environment, i. e. of the patient’s condition and the data associated with it, resulting in a reward or a penalty for the agent based on whether or not the chosen action led to a favorable outcome for the patient. In that sense, the agent can abstract a policy either from rewards or state-action pairs that drives action selection, for example the agent may learn that increasing doses of chemotherapy are associated with increased anti-tumor affect, but also increased toxicity.” (Lines 113-122). This general framework is the underlying principle of most of the referenced literature. We believe this may aid the reader in better understanding the basic vocabulary of RL and infer these abstract concepts to clinical examples given in paragraph 3.